# Reach, Acceptability, and Sustainability of the Native Changing High-Risk Alcohol Use and Increasing Contraception Effectiveness Study (CHOICES) Intervention: A Qualitative Evaluation of an Alcohol-Exposed Pregnancy Prevention Program

**DOI:** 10.3390/ijerph21030266

**Published:** 2024-02-26

**Authors:** Sara M. London, Jessica D. Hanson, Michelle Sarche, Kyra Oziel, Dedra Buchwald

**Affiliations:** 1Institute for Research and Education to Advance Community Health, Washington State University, 1100 Olive Way, Suite 1200, Seattle, WA 98101, USA; kyra.oziel@wsu.edu (K.O.); dedra.buchwald@wsu.edu (D.B.); 2Department of Applied Human Sciences, University of Minnesota Duluth, 1216 Ordean Court, Duluth, MN 55812, USA; jdhanson@d.umn.edu; 3Centers for American Indian and Alaska Native Health, Colorado School of Public Health, University of Colorado Anschutz Medical Campus, 13055 E. 17th Avenue, Aurora, CO 80045, USA; michelle.sarche@cuanschutz.edu

**Keywords:** alcohol-exposed pregnancy prevention, American Indian, satisfaction, reach, acceptability, sustainability, COVID-19 pandemic

## Abstract

American Indian (AI) women are at risk of alcohol-exposed pregnancy (AEP) due to the higher prevalence of alcohol use disorders (AUDs) and risky drinking. The Native Changing High-Risk Alcohol Use and Increasing Contraception Effectiveness Study (Native CHOICES) was implemented in partnership with a Northern Plains Tribal community to address the effectiveness of a brief, motivational interviewing-based intervention to reduce AEP risk among adult AI women. A subgroup of the participants shared their perspectives in a qualitative interview conducted following the completion of the six-month post-baseline data collection. These interviews solicited participant perspectives on the Native CHOICES intervention and its satisfaction, reach, acceptability, and sustainability. The participants were delighted with Native CHOICES, felt the intervention helped them learn about AEP prevention and goal setting, learned valuable lessons, and believed Native CHOICES would be well-received by other women in their community and should be continued. The participants also shared how the COVID-19 pandemic affected their choices about drinking and birth control. The findings showed the receptivity to and acceptance of Native CHOICES among AI women. The interview findings offered a glimpse into the effectiveness of Native CHOICES and how it contributed to participants making healthier choices surrounding drinking and sexual health.

## 1. Introduction

Fetal alcohol spectrum disorders (FASDs) are a serious public health problem and are 100% preventable. Caused by alcohol use during pregnancy, FASDs are a continuum of behavioral and cognitive impairments with lifelong negative consequences [1,2,3]. Even moderate alcohol use during pregnancy can affect fetal growth and developmental outcomes, and many children with FASDs never receive a formal diagnosis [4]. Among women of childbearing age, pre-conceptual alcohol use disorders (AUDs), and risky drinking (i.e., binge and heavy) prior to pregnancy are among the most substantial risk factors for an alcohol-exposed pregnancy (AEP) that can lead to FASDs in their children [5,6].

Data on alcohol use during pregnancy among American Indian and Alaska Native (AI/AN) women are mixed. Overall, AI/AN women have a higher prevalence of AUDs than White women (7.9% vs. 4.9%) [7] and are more likely to engage in risky drinking during pregnancy [8,9]. In some studies, AI/AN women report having high rates of heavy alcohol consumption and binge drinking; however, others [10,11] found AI/AN women reported alcohol abstinence rates similar to women in the general population. AI/AN people often face barriers to accessing AUD treatment [12,13] due to residing in rural areas and the scarcity of qualified providers [14].

Many interventions to prevent FASDs aim to reduce AUDs and risky drinking among pregnant women. However, a growing body of research indicates that AEP prevention must begin before conception [10,15]. Unfortunately, the evidence base for such preconception prevention among AI/AN women is meager. A review of thirteen studies with evidence-based approaches for the reduction in alcohol consumption by AI/AN women who are pregnant or of reproductive age highlighted limitations such as nonrandom samples, lack of a control group comparison, and reliance on self-reported alcohol and contraceptive use [16]. One promising approach for preconception AEP prevention is the Changing High-risk Alcohol Use and Increasing Contraception Effectiveness Study (CHOICES) [17], which combines motivational interviewing with contraceptive counseling. In a randomized trial among non-AI/AN women, those who received CHOICES reduced their risk of AEP by an average of 16.6% across three-, six-, and nine-month follow-ups compared to women who received usual care [17]. CHOICES has been adapted for and implemented with AI women from a Northern Plains Tribal community [18]. Still, evidence of its effectiveness, as demonstrated by a randomized controlled trial (RCT), awaits the results of the Native CHOICES study we have just completed. 

The Native CHOICES RCT is described in detail elsewhere [18]. The researchers had a longstanding relationship with the community site, and they utilized community-based participatory research throughout the study. They had a community action board (CAB) comprising community leaders, Elders, and Indian Health Service providers who were involved in every aspect of the study from design to dissemination; the results and final paper were reviewed by the CAB [18]. Briefly, AI women aged 18–44 years old who were at risk for an AEP, because they were engaged in heavy or binge drinking, able to become pregnant, and not using effective birth control, were recruited to participate, and randomly assigned one-to-one to an intervention or control group. The Native CHOICES intervention comprised two motivational interviewing sessions, optional contraceptive counseling visit with a medical provider, and optional consistent contact through supportive electronic messaging to encourage participation. Data were collected at baseline and six weeks, three months, and six months post-baseline to assess intervention effectiveness. We assessed alcohol consumption by both self-report and an alcohol biomarker (phosphatidylethanol); contraceptive use, smoking, and depressive symptoms were assessed by self-report. Native CHOICES aimed to reduce AEP risk by reducing risky drinking or increasing effective contraception use, and ideally both [18]. This paper reports on the results of qualitative interviews conducted following the final intervention follow-up data collection at six months post-baseline. Interviews solicited participant perspectives on their satisfaction with Native CHOICES and its acceptability, reach, sustainability, and future directions. 

## 2. Materials and Methods

This study was approved by the Washington State University Institutional Review Board and followed all protocols for Tribal review. All women enrolled in the RCT were eligible to participate in a qualitative interview after completing the six-month post-intervention follow-up data collection. The initial informed consent also permitted the research team to contact women for subsequent other studies, which they could accept or decline. Women who agreed to participate were re-consented for the qualitative interview. Both verbal and signed consents were approved to accommodate participant preferences during the COVID-19 pandemic. Qualitative interviews solicited participants’ reflections on their satisfaction with Native CHOICES, perceptions about the intervention’s reach, acceptability, and sustainability, and recommendations for intervention improvement and expansion. Besides providing quotes on participants’ qualitative responses below, any additional information on the trial, such as demographic information or the results of the study, will be provided in separate manuscripts.

### 2.1. Data Collection

Interview questions assessed satisfaction with the Native CHOICES intervention. Women were asked, “What did you learn from participating in CHOICES?”, “What did you like best about CHOICES?”, and “What did you like least about CHOICES?”. A follow-up question asked, “What would you change to make CHOICES better?”. Other questions probed the acceptability of the intervention, for example, “How well do you feel CHOICES fits with your community’s Native value system?”, and “Do you think taking part in CHOICES would be appealing to other women you know?”. To assess the reach of the intervention, women were asked, “How can we best reach other people in the community to discuss risky drinking and effective birth control?”, and “How could the women in your circle benefit from greater access to this intervention?”. Finally, the sustainability of the intervention was evaluated by asking, “How can we keep CHOICES going?”.

Interviews lasting 30 to 60 min were conducted remotely over the telephone or by Zoom. Interviews were audio recorded and transcribed by a Health Insurance Portability and Accountability Act (HIPAA)-compliant and encrypted online transcription service. Each participant received USD 50 for participating.

### 2.2. Data Analysis

Through deductive thematic analysis [19], two coders identified themes based on the existing categorical responses to questions about the intervention’s reach, acceptability, and sustainability. Using Dedoose software package [20], coders analyzed transcripts separately to identify patterns and create codes, met weekly to discuss the themes they identified in the first three transcripts, and created a codebook. Coders then used the codebook to code subsequent transcripts and met to discuss the findings to create a document with excerpts that represented the themes. Double-coding a random transcripts sample, the coders reached 100% of Cohen’s kappa coefficient for interrater reliability [21]. To protect participant privacy and the confidentiality of their data, each participant’s transcript was also assigned a pseudonym. 

## 3. Results

While 404 individuals were enrolled in the Native CHOICES intervention, a subgroup of *n* = 40 completed the qualitative interviews. Our results are organized by the major foci of our evaluation, namely: reach, acceptability, and sustainability. For each, we present exemplary quotes. Additional themes examine the impact of the pandemic on women’s choices about drinking, birth control, and sexual health. 

### 3.1. Reach

Within the broad theme of “reach,” participants shared their motivation to take part in Native CHOICES, how it helped them, and what they learned from participating in Native CHOICES. 

Motivation to take part in Native CHOICES. The participants shared that their primary motivations for taking part in Native CHOICES were to understand their drinking habits, become empowered to make healthy choices, reduce their risk of an AEP, and learn more about themselves and their habits. 

“I learned about your program through Facebook and wanted to look at myself with the alcohol choices and stuff, like, whatever options. Kind of like basically wanting to see what I was going through, I wanted to see it through your study. See what I was doing, what led my life to alcohol”.(Amy)

Participants also shared the common theme of being motivated to join Native CHOICES out of curiosity about what the program entailed. Either they heard about Native CHOICES from a friend and wanted to participate themselves, or they were intrigued by other recruitment efforts. 

“At first, one of my friends recommended me to come over here because she just got done with the study, and she said that it was a really good learning experience for her as well as research study. And it intrigued me to want to come do it, so I came this way”.(Sarah)

How Native CHOICES helped. Several participants addressed how learning about AEP risk helped give them tools to be more health conscious, which allowed them to quit unhealthy habits and engage in new, healthier ones.

“It did. And it actually helped me to really be conscious about my health as a woman and, I guess, with contraception. And it really helped me to open up and think about other family members and how it’s not a subject that everybody talks about”.(Angela)

What participants learned. The participants valued Native CHOICES because it provided new insights, tools, and lessons for addressing their sexual habits, learning about sexual health, setting goals to change their drinking habits, and generally becoming aware of their drinking habits. Many shared their success in this regard.

“There was a lot going on. A lot of people drink alcohol, use cigarettes, have unprotected sex because we’re not making a rational thinking. And there’s a lot more that comes from that than just HIV and STDs. There’s a lot more to it. So, we just get more information about certain things, and a lot of it pertained to women. So, it’s kind of like just a women’s program”.(Erin)

Another participant discussed how she never thought her drinking was risky, but through Native CHOICES, she realized it was.

“I’ve also learned about risky drinking. You really don’t think about it until we talk about it. Because I didn’t think I was a risky drinker and I am. I have never thought about it. I thought alcohol is just something fun to do. But then it’s also risky for yourself as a woman and your reproduction”.(Maria)

Another reflected how she looked at herself differently when it came to drinking following her participation in Native CHOICES.

“I started looking at myself and seeing myself through a different perspective of my alcohol use, and I was trying to slow down, but I just never really thought about (it). After I started answering the questions and stuff, I started seeing myself in a different way and it helped me want to change”.(Amy)

One participant valued Native CHOICES for what they learned about sexual health, including how to obtain birth control, the prevalence and prevention of sexually transmitted infections, and how to prevent AEP. 

“Well, I learned that I wasn’t on birth control and that I was a heavy drinker. And then it is true, you should just start being protective. And I did get on birth control after that, though. So, I’ve been on birth control, so it helped me because I can’t really handle it myself. So, I’m glad that I got on birth control because if I happen to get pregnant, I wouldn’t know how I would be able to take care of the kid”.(Talia)

Reflecting on aspects of Native CHOICES that supported their insights and behavioral changes, participants felt that the client workbook was particularly helpful. Some shared that the workbooks and lessons they utilized in the Native CHOICES intervention guided them to become more health conscious, get on the right path, gain a new perspective, and address other health priorities, for example, quitting smoking. The participants also identified goal setting as an important aspect of Native CHOICES that led to their behavioral changes.

“I remember in CHOICES I made goals, and I said I wanted to stay sober and be sober, and that’s over 10 months ago now and I’m still sober and loving life and I’m still not pregnant. But I still don’t believe in birth control, so I’m not even using it. But I am loving the sober life and Native CHOICES help me set those goals to make me want to stay sober. And here I am, still sober 10 months later”.(Amanda)

Along with goal-setting specific to drinking, participants also shared their appreciation for setting realistic goals to move forward in life more generally.

“I’d just like to thank your program for helping me look at myself in a different way and those questions that I answered really helped me, and I’m still pushing forward with my goals. And it’s a struggle but I’m making it”.(Amy)

Talking to Native CHOICES staff and having permission to be vulnerable were also noted as helpful. The participants felt comfortable sharing their experiences because they knew the information was confidential, and they trusted and valued the Native CHOICES staff. 

“I like the staff. You guys are all very friendly and helpful. Like I said, helped with referrals to the women hospitals and gave us pamphlets and information. And it was really nice. It was just a nice atmosphere. Nice to go in and visit with people. We don’t get to see you guys every day. So, it was nice to just go visit someone different”.(Erin)

### 3.2. Acceptability

Participants believed Native CHOICES would be acceptable to other women in their community because of its focus on the effects of alcohol and drug use during pregnancy. The participants stated that other women should have the opportunity to learn about this topic and that young women in their lives and family members would find Native CHOICES appealing. Most participants also noted Native CHOICES fit traditional AI values by promoting well-being and fostering respect for themselves and others.

“I think it fits in well, because it promotes—I would think for women to want respect to theirselves more and respect others and value their virtues, because a lot of people don’t even really know. And that’s another thing. With this program they teach you that, so that would probably be a self-esteem booster if they would know—to be knowing their self-worth and to respect theirself”.(Clarisse)

Participants shared what they liked best and least about the intervention. They enjoyed learning about alcohol, reflecting on their drinking habits, and understanding how alcohol impacted pregnancy. Women also appreciated learning about different types of birth control and how to use them, as well as the close and confidential relationship with the Native CHOICES interventionist, with whom they felt valued and safe. Only a few women articulated an aspect of Native CHOICES they liked least; among those who did, the amount of time the intervention took and learning painful truths about their alcohol habits were noted.

### 3.3. Sustainability

Participants shared recommendations for sustaining and expanding the intervention. They believed it would benefit younger women and recommended more advertisements and outreach in the community. They also shared that incentives were essential and should always be included. 

One participant shared that her initial reason for joining was receiving an incentive to help her with life costs, but she later appreciated what she learned.

“So for me, I like that it helped me see my drinking habits and stuff. But I think for them—and the only reason why I, honestly, did participate was because of the compensation. But I think the compensation could help them with gas or whatever and make them more aware of self—Because I understand some people—but I feel like when they come in just to get the money, and then they’re like, ‘Oh, well, I actually learned something.’ You know what I mean? A little bit. So I don’t know. I feel like—Yeah, that was totally me”.(Lucy)

One participant shared information she learned—about sexual health, birth control, the prevalence and prevention of sexually transmitted infections, and how to prevent AEP—with her nieces and pointed out the importance of continuing Native CHOICES for her younger family members. 

“Yeah. I got nieces and stuff that I show those information stuff to. They would like to do stuff like that, too. They’re young and doing that stuff now too”.(Miranda)

Participants shared recommendations for recruiting and advertising in their community. Women recommended using Facebook and more traditional venues like the radio, newspaper ads, flyers posted on bulletin boards around buildings in the community, such as the grocery store and post office, and passing out flyers to community members. They also recommended sharing information about Native CHOICES in more traditional settings, such as community Talking Circles. One participant shared the importance of recruiting participants through word of mouth and simply talking to women in the community through different mediums to foster comfortability and an environment where they can feel more open to learning and participating. 

“I think I’ve heard a radio ad. I think that’s a great way to put it out there. And I also think even just talking about it. Like having a recorded session with people who are open to talk about it and advertising that, whether it’s a Facebook Live or a radio portion, I think when people see someone talking about something so comfortably, then they’re open to join in”.(Maria)

Other suggestions were to collaborate with community-based programs serving individuals who may be interested in Native CHOICES, such as family violence programs, Community Health Representative programs, women’s clinics, and the Indian Health Service.

### 3.4. Impact of the Pandemic

Participants reflected on the ways in which the pandemic affected their drinking habits and access to birth control. Many drank less during the pandemic due to social distancing, travel restrictions, and fear of contracting COVID-19, which restricted socializing. However, a few women drank more, a few quit altogether, and a few reported the pandemic did not impact their drinking habits at all.

“Actually, I quit during COVID because sharing alcohol and just like transferring it from your saliva, it’s hitting the bottom of the bottle. And just, no”. (Claire)

Among women who drank more during the pandemic, isolation, lack of structure, more free time, and travel restrictions were noted as contributors.

“It (drinking) was a lot worse. There was nothing to do, with no jobs hiring or anything. And I don’t know, all my friends are doing it, too. And I didn’t want to be alone. And I started falling with everybody else”.(Amy)

With respect to birth control, many participants reported their use was not affected. However, a few reported poor access to birth control due to stay-at-home orders, closed clinics and hospitals, or overcrowding at clinics and hospitals.

“The clinics were closed—Yeah, there was no work. There were times I had gotten sick, but I couldn’t go to the hospital to get birth control and I know the programs weren’t open and everyone was on leave for the pandemic. Everybody got to stay home”.(Amy)

Conversely, some women reported using birth control due to drinking behaviors during the pandemic and wanting to protect themselves.

“So for me, during COVID, personally, I noticed I’ve started drinking a lot more alcohol. I don’t know if it was freedom or time or being quarantined. It kind of felt like there was not very much structure during COVID or during quarantine. But I definitely was more aware of using birth control because I knew I was drinking more alcohol”. (Angela)

## 4. Discussion

These interviews aimed to understand the reach, acceptability, and sustainability of the Native CHOICES program among participants and explore their perceptions of the pandemic’s effects on their drinking habits and birth control access and use. There are some limitations to this study, including the sample being from a relatively small geographic area in the Great Plains of the United States. Therefore, the findings from this study cannot be generalized to American Indian women overall or even American Indian women from all tribal communities in the Great Plains. Also, as with any qualitative study, there is the potential for analysis bias, recall error, reactivity of the participant to the interviewer, and self-serving responses [22].

Overall, participants appeared satisfied with their study experiences. Native CHOICES promoted their understanding of AEP prevention and helped them set and achieve their health goals. Other studies of CHOICES reduced the risk of AEP among participants and enhanced well-being. In one study, Parrish et al. (2019) adopted a version of CHOICES and found their study feasible and acceptable to participants and reduced substance-exposed pregnancy [23]. This intervention was comprehensible to participants and was neither overly time intensive nor did it exceed the planned duration. The participants also valued the confidentiality of the intervention, and all of them would recommend the intervention to a friend. The participants in this study were moderately to highly satisfied with the services, enjoyed the confidentiality of the intervention, and would recommend the intervention to their friends and family.

Overall, participants enjoyed being part of the study, felt safe, and appreciated the staff and the lasting relationships they built with the CHOICES staff. The program fits with their traditional values by promoting respect and providing a safe space for healing. The literature underscores the importance of Native-led research, as it builds trust with underrepresented communities where, historically, many were harmed [24,25,26]. Our study illuminates the importance of addressing historical harm and how building trust and creating a safe and confidential environment can foster satisfaction with research experiences. 

Several other existing interventions to reduce alcohol consumption have been culturally adapted for Native communities. In a study on the feasibility and acceptability of a culturally tailored smartphone app to reduce alcohol use, AI/AN participants appreciated the confidentiality and privacy of the intervention because anonymity reduced the stigma around drinking [27]. Similarly to Native CHOICES, participants were enthusiastic about the intervention and believed it would benefit younger generations. Our participants also valued learning about STIs, birth control, and sexual health. Similar findings were highlighted in a study of a culturally adapted virtual teen pregnancy prevention and sexual reproductive health intervention. Their AI/AN participants acknowledged the importance and value of talking about sexual and reproductive health, a topic not often addressed in their families and communities [28]. These studies, among others, align with views regarding the feasibility and acceptability of Native CHOICES. This cross-cutting theme demonstrates the value of culturally specific interventions and offers insight into future adaptations of interventions designed for Native communities. 

Participants in Native CHOICES agreed the intervention would appeal to other women in their social groups, specifically younger generations, and they wished to see the benefits passed onto others. Taken together, our findings and the extant literature highlight the need for prevention efforts such as Native CHOICES to serve younger generations [29,30]. Lastly, participants provided community-specific ways to sustain the intervention, for example, by partnering with other community programs serving individuals who have a need for or interest in Native CHOICES. This feedback demonstrated how Native CHOICES holds promise for supporting Native women with the desire to learn about AEP risk reduction beyond the end of our study.

## 5. Conclusions

Through interviews conducted after completion of RCT data collection exploring the reach, acceptability, and sustainability of Native CHOICES, participants in this study seemed to have a high degree of satisfaction with the Native CHOICES intervention. Native CHOICES helped them make healthier and more positive choices, value learning about their drinking habits and sexual health, and appreciate the safety and comfort they experienced working with the Native CHOICES staff. Women felt Native CHOICES fit with their traditional AI values and practices and believed it should continue to be offered. They were enthusiastic about offering the program to younger women and family members to promote well-being in their community. The participants highly accepted and appreciated the motivational interviewing, contraceptive counseling, and supportive staff. This study is an important example of how CHOICES can be utilized within a culturally specific setting by incorporating Native staff and values and building trusting relationships to reach their participants. These adaptations increased participant receptivity to learning and making healthier choices about drinking and contraception, ultimately meeting the Native CHOICES goal of reducing the risk of AEP among AI women. 

## Data Availability

Data presented in this study are under the control and ownership of the tribal nation and oversight body that partnered in this research. They may be requested from the principal investigator of this study (D.B.), but it is up to the tribal nation and oversight body to decide their availability.

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
