# Peer review of "Reach, Acceptability, and Sustainability of the Native Changing High-Risk Alcohol Use and Increasing Contraception Effectiveness Study (CHOICES) Intervention: A Qualitative Evaluation of an Alcohol-Exposed Pregnancy Prevention Program"

_ijerph, 2024, doi:10.3390/ijerph21030266_

Round 1

Reviewer 1 Report

Comments and Suggestions for Authors

I would like to thank the authors for their work. This is an interesting paper, which aims to report the results of qualitative interviews conducted following the final intervention follow-up of the Native CHOICES Randomized Controlled Trial. 

Before the publication I have some minor suggestions.

1) Title: I suggest to adopt a more informative title, including a bit of information on the nature of the trial

2) Results: even this is not a quantitative study, some informations on the included sample would allow a more in-depth comprehension of the results (i.e. age, education). Maybe using a table? Of course, this is just a suggestion

3) Results: The quoted sentences of the participants could be shown in italics to improve the readability of the paper.

4) Results: lines 317-318. This sentence is graphically set up as if it were a quote from a participant, but it looks like a typo

5) Discussion: lines 345-346. "Overall, participants were very satisfied with their study experiences". Since satisfaction was not directly measured quantitatively, I suggest "dampening" this sentence, as it may results as speculative

6) Discussion: I suggest to add the limitations of this study at the end of this section

7) Conclusions: lines 391-392. "...participants in this study expressed a high degree of satisfaction with the Native CHOICES intervention." Same observation as point 5

Reviewer 2 Report

Comments and Suggestions for Authors

How many women participated in the program?

How did you achieve saturation?

Did the authors consider using quantitative measures of satisfaction, reach, acceptability, and sustainability?

Why were programmatic staff excluded from these measures? While the direct impact certainly is relevant to participants, information from the program staff may better assess reach and sustainability, for example.

Was there a theoretical approach to the analysis? This seems like an important point that is missing.

Some quantitative details may help substantiate the conclusions. For example, how many of the total participants agreed or mentioned the items reported as findings.
